# OpenReview forum: "Unified Language Model Alignment with Demonstration and Point-wise Human Preference"
_ICLR.cc/2024/Conference — Submitted to ICLR 2024_

### Official Review · Reviewer_33oF · 2023-10-28

**Soundness:** 2 fair
**Presentation:** 2 fair
**Contribution:** 2 fair
**Rating:** 5
**Confidence:** 3

**Summary:**

This paper presents a unified framework that integrates the two traditionally separate processes in LLM alignment: SFT on demonstration data and preference learning on preference data. The framework is structured for point-wise preference learning, considering the intrinsic characteristics of real-world preference data distribution. Specifically, the authors treat the positive and negative samples differently, applying SFT loss to the former and adding an additional KL regularizer for the latter. They also justify this formulation via gradient analysis and comparison with DPO. Since the method enhances learning from the positive samples, the authors extend the Anthropic's Helpful and Harmless dataset by refining the positive responses using GPT-4, thus boosting their method with high-quality data.

**Strengths:**

Reward modeling is a crucial and challenging part in LLM alignment. Conventionally, preference data is collected through rankings and used as preferred–dispreferred pairs for learning, as the human-annotated scalar scores on individual samples can be uncalibrated and noisy. However, the ranking-based reward model may fail to impose correct penalty, since it is trained based on binary relative signals, potentially compromising its precision on individual samples. In this case, I agree with the authors that pair-wise RM may inadequately capture the nuances of real-world preference data distribution, especially on the data where preferences are obviously polarized and scoring quality against specific criteria is unambiguous. Therefore, I think it is important to explore the point-wise RM for better preference learning in LLM alignment.

**Weaknesses:**

While preference learning from pair-wise data is challenging (as I briefly discussed above), it still applies to most cases in the real world. For example, toxicity is not a strictly binary metric as we can categorize samples to be _toxic_, _very toxic_, or just _pose risks of toxic content generation_ [1]. Also for verifiability, there can be labels such as _unhedged correct_, _hedged correct_, and _uninformative_. So I don’t think the authors made a convincing argument regarding the superiority of their point-wise preference learning over the pair-wise methods. In fact, the binary signals could result in significant information loss, since the learning can only capture the data polarity, omitting the nuanced levels present in practice.

Additionally, the paper lacks empirical analysis with limited experimental results to justify the design of each component in Equation (9). It is hard to interpret how the win rates evaluated by GPT-4 correlate with human judgment or the actual quality. For example, if I understand it correctly, the baseline to compare against is the chosen answer in the dataset, which can be considered as the golden samples for preference learning. This makes the numbers of win-rates in Tables 1&2 somewhat weird and vague since there should be a big proportion of tie cases as indicated in previous works [2]. It’d help to report on metrics that are consistent with existing works for clear and interpretable result comparison. It is also important to extend the evaluation to other benchmarks, _e.g._, RealToxicityPrompts [1], to compare their effectiveness at least in the domain of harmlessness.

---
[1] Gehman et al. Evaluating Neural Toxic Degeneration in Language Models. Findings of EMNLP 2020

[2] Rafailov et al. Direct Preference Optimization: Your Language Model is Secretly a Reward Model. 2023

**Questions:**

A. Could the authors elaborate more on the design of point-wise preference learning, particularly regarding harmlessness and helpfulness? For example, how to deal with potential information loss when simplifing the label to be strictly binary?

B. The win-rates, especially in Golden HH, are close to $100$%. Could you elaborate on the reasons behind these statistics and also provide information on the corresponding lose- and tie-rates?

C. How did the authors obtain and evaluate the baseline results? For example, there isn’t an official implementation of DPO, how did the authors ensure that their version of DPO is consistent with the original one, and how does their result align with the reported one in the DPO paper?

---

> ### Author Response · Authors · 2023-11-19
> **Author Response (Part 1)**
>
> Thank you for the detailed review! In the following, we will address your questions and give clarification on various aspects of this paper.
>
> **Q1:** "the binary signals could result in significant information loss, since the learning can only capture the data polarity, omitting the nuanced levels present in practice"
>
> **A1:** Admittedly, as you have pointed out, many datasets on toxicity and verifiability are not binary labelled. Our original statement regarding binary labeling on toxicity and verifiability in Section 4.1 is only meant for providing an example to illustrate the limitation of pair-wise preference learning methods on point-wise preference datasets.
>
> We shall clarify that such a statement does not mean that our proposed ULMA is restricted to binary point-wise datasets. In fact, **we do not necessarily need to transform continuous labels into binary labels when applying ULMA**. The core concept of ULMA, i.e., using a hybrid objective formulation to treat the high-quality demonstration data and possibly noisy preference data differently, is capable of generalizing to continuous labels to characterize the different levels.
>
> Specifically, for these datasets, there is no direct separation of positive or negative samples as in the binary case. To apply ULMA, we can specify some sample as high quality data (e.g., the most helpful or harmless ones) and treat these high quality samples as "positive" demonstration data. Then we can adopt a hybrid loss by combining the SFT loss on these demonstration data with the original loss (e.g., MSE) on preference data.
>
> In the revised manuscript, we add the above clarification and also provide a concrete example on the *red-team* dataset with continuous labels. *Red-team* is a point-wise dataset on LLM's robustness to red teaming attacks. In this dataset, each sample has a score ranging from 0 to 4, which is rated by human and indicates how successful the attack is to the LLM (in Likert scale; an attack with a higher level is more successful). *Red-team* is very similar to your mentioned datasets on toxic content generation or verifiability. However, this dataset only has a single sample for each prompt, hence no comparison can be made and pair-wise methods such as RLHF and DPO are inapplicable; we can only apply point-wise methods such as our proposed point-wise DPO.
>
> To further improve the performance of point-wise DPO on *red-team*, inspired by the ULMA framework for binary cases, we treat the samples of level 0 as the high quality demonstration data. Then we introduce a hybrid loss by adding SFT loss of the samples rated 0 to the original MSE loss of all samples. As presented in Table 1, empirical results show that our proposed ULMA outperforms other methods on *red-team*. Please refer to the revised manuscript for more detailed description on the dataset and empirical results.
>
> **Q2:** "lacks empirical analysis with limited experimental results to justify the design of each component"
>
> **A2:** In the revised manuscript, we have conducted an ablation study to verify the design of the hybrid objective formulation in ULMA. Specifically, we first use positive samples as demonstration data to compare ULMA with point-wise DPO (which adopts KL regularization for these samples) to evaluate the effectiveness of ULMA for learning from high quality (positive) demonstration data. Then we use negative samples as dispreferred demonstration data to compare ULMA with the counterpart algorithm without KL regularization (i.e., Unlearning) to evaluate the ability of ULMA for learning from possibly noisy (negative) preference data. As presented in Table 3, the results show that ULMA can better exploit the high quality positive samples by applying SFT loss to these samples, while preventing from overfitting to possibly noisy negative samples via adding regularization to negative samples. Please refer to the revised manuscript for more detailed empirical results.

---

> ### Author Response · Authors · 2023-11-19
> **Author Response (Part 2)**
>
> **Q3:** "the numbers of win-rates in Tables 1 and 2 are somewhat weird and vague... The win-rates, especially in Golden HH, are close to 100 percents"
>
> **A3:** The numbers reported in Table 1 and 2 are actually the **win \& tie** rate [1, 2] of the harmful metric. We had made a typo in the original captions of Table 1 and 2. Sorry for causing your confusion! In the revised manuscript, beyond the win \& tie rate presented in Table 1 and 2, we also report the detailed win, tie and loss rates respectively in Figure 1 and Figure 2 in Appendix D.
>
> We also note that it is reasonable that the win \& tie rates on
> *Golden HH* are close to 100\%. Here we explain the reason. Since *Golden HH* is constructed based on *HH*, to make the results on *Golden HH* and *HH* comparable, the generated answers are **compared to the chosen sample in the original *HH* dataset** when evaluating the harmful score on *Golden HH*. However, the quality of the chosen samples in *HH* is not very high. When employing SFT on *Golden HH* where the new chosen data are of much higher quality, the answers generated on *Golden HH* is very likely to be at least no worse than the original chosen data in *HH*. As a consequence, the win \& tie rates on *Golden HH* are close to 100\%. Note that the above phenomenon actually implies that the quality of the chosen samples in *Golden HH* is much higher than that in *HH* (also see Table 5 in Appendix C for more detailed comparison of both datasets).
>
> **Q4:** "It is also important to extend the evaluation to other benchmarks"
>
> **A4:** We have added two more datasets *QA-feedback* and *red-team* in the revised manuscript. The first dataset *QA-feedback* is a point-wise QA dataset with binary labels, in which the perplexity and the helpful score are evaluated. The second dataset *red-team* is a point-wise dataset with continuous labels on robustness of LLMs to red teaming attacks, in which we report the perplexity and the harmful score. The empirical results are presented in Table 1 and Table 2 in the updated manuscript, from which we also observe that our proposed ULMA method outperforms other compared methods on both datasets. Please refer to the manuscript for empirical results and detailed discussions.
>
> **Q5:** "there isn’t an official implementation of DPO... how does their result align with the reported one in the DPO paper"
>
> **A5:** Indeed, in our experiments, we had followed the official implelentation of DPO in https://github.com/eric-mitchell/direct-preference-optimization. The reported numbers are a bit different from those in the DPO's paper, because we adopt the harmful score as the metric on the *HH* dataset, which is different from the helpful score adopted by the DPO's paper.
>
> [1] Duan et al. BOTCHAT: Evaluating LLMs’ Capabilities of Having Multi-Turn Dialogues. ArXiv, abs/2310.13650, 2023.
>
> [2] Xu et al. Superclue: A Comprehensive Chinese Large Language Model Benchmark. ArXiv, abs/2307.15020, 2023.

---

> > ### Comment · Reviewer_33oF · 2023-11-20
> >
> > Thanks for clarifying my concerns and adding new results!
> >
> > My additional comments:
> > - I noticed that the point-wise DPO presents comparable performance as ULMA. As the authors discussed, ULMA is better since it removes the regularization to enhance high-quality data learning. But from my understanding, the improvement actually depends on the quality of the positive samples and how much they can deviate from the reference model distribution. In the paper, the authors revise the HH data to be Golden HH to have better positive samples. But I am wondering whether this is generalizable to cases where we only have relatively high-quality samples or limited high-quality samples. For example, samples rated 0 by different people may have various qualities when the ratings are subjective. In this case, how would ULMA be impacted by the positive data quality? And how would it perform if given unbalanced positive & negative samples?
> > -  In Table 3, the GPT4 evaluation score of Unlearning on QA-feedback is $<0.5$. Does this mean that it makes things worse compared with the reference model (training starting point)?

---

> > > ### Author Response · Authors · 2023-11-21
> > > **Response to your additional comments**
> > >
> > > Thank you for your valuable feedbacks! We would like to address your additional comments as follows.
> > >
> > > **Q1:** "whether this is generalizable to cases where we only have relatively high-quality samples or limited high-quality samples."
> > >
> > > **A1:** Admittedly, the performance gain of ULMA compared to point-wise DPO increases as the quality of the chosen data gets higher. However, it does not mean that ULMA is restricted to samples of very high quality; the gain still exists on relatively high-quality samples. To empirically verify this point, we construct a new dataset with relatively high-quality samples by replacing **half of the chosen data** in *HH* by *Golden HH*, and then compare the performance of point-wise DPO and ULMA on the new dataset. We find that ULMA (win \& tie rate 0.93 for the harmful metric) still outperforms point-wise DPO (win \& tie rate 0.89 for the harmful metric) on such dataset, which is consistent with the previous results on *HH* and *Golden HH* in Table 1.
> > >
> > > As for the case of limited positive samples, we just conducted a new experiment on the red team dataset by discarding two thirds of the positive samples. In this way, the ratio of the positive samples drops from 42.4\% to 19.6\%, and we find that ULMA (win\&tie rate 0.89 for the harmful metric) still outperforms the point-wise DPO (win\&tie rate 0.88 for the harmful metric). The above results show that ULMA still has advantage with relatively high-quality samples limited high-quality samples. We will add the above results and discussions in the revision.
> > >
> > > **Q2:**: "Does this mean that Unlearning makes things worse compared with the reference model (training starting point)"
> > >
> > > **A2:**: Indeed, in Table 3, the unlearning method performs worse than the reference model. Here we explain the reason. For the purpose of ablation study, in our experiments, the unlearning method does not include pre-training via SFT; it only uses negative samples to fine-tune the pre-trained model. In other words, unlearning does not use high-quality positive samples in preference dataset for model alignment. As the negative samples in the datasets are noisy and of relatively low quality, unlearning without regularization may suffer from ***catastrophic unlearning*** [2, Fig 3 of 1]. The undesirable performance of unlearning on possibly noisy samples has actually been supported by our ablation study, in which negative DPO improved the performance by adding regularization on the noisy negative samples compared to unlearning. This actually motivates us to impose regularization on relatively low quality data as in ULMA.
> > >
> > >
> > > [1] Lu, Ximing, et al. "Quark: Controllable text generation with reinforced unlearning." Advances in neural information processing systems 35 (2022): 27591-27609.
> > >
> > > [2] Nguyen, Quoc Phong, Bryan Kian Hsiang Low, and Patrick Jaillet. "Variational bayesian unlearning." Advances in Neural Information Processing Systems 33 (2020): 16025-16036.

---

> > > > ### Comment · Reviewer_33oF · 2023-11-23
> > > >
> > > > Thanks for your response. I appreciate the additional experiments and explanations. I will maintain my scores as the updates have not fully addressed my initial concerns about the impacts of key components, such as the regularization in the proposed method.  However, I believe it is important and promising to explore point-wise preference modeling. And it would be beneficial to have a more detailed analysis of how the key components influence model outputs, like the different effects of regularization on language quality and toxicity, as shown in Fig 3 of [1].
> > > >
> > > > [1] Lu et al. Quark: Controllable text generation with reinforced unlearning. NeurIPS 2022

---

> > > > > ### Author Response · Authors · 2023-11-23
> > > > > **Response to your additional comment**
> > > > >
> > > > > We are sincerely grateful for your continued engagement and thoughtful feedback on our manuscript. We understand your concerns regarding the impacts of key components like regularization and appreciate your suggestion for a more detailed analysis.
> > > > >
> > > > > We would like to clarify that we have not conducted an analysis of the regularization component's impact in our original manuscript, since the substantial body of existing literature [1,2,3] has extensively studied how the regularization term making trade-offs between multiple rewards, such as the language quality and toxicity as you mentioned. Our expectation that the findings reported in these prior works would extend to our method led us to focus on the novel aspects of our contribution.
> > > > >
> > > > > However, we recognize the importance of empirical evidence to support our claims, and in light of your feedback, we are committed to conducting a thorough analysis of how regularization specifically influences the outputs of our model. This will include a detailed Pareto frontier analysis, that will illustrate the nuanced effects of regularization on both language quality and helpful / harmful metrics. This additional experiment will be carefully designed to addresses your valid concerns.
> > > > >
> > > > > We would also like to highlight that the primary contribution of our paper is the introduction of a unified framework that adeptly learns from both demonstration and point-wise preference datasets. This versatile framework enables the use of SFT and various preference learning methods, including the application of different regularization terms. The proposed ULMA method empirically achieves significant performance improvements across a spectrum of data quality scenarios.
> > > > >
> > > > > We thank you once again for guiding us towards enriching our study, and we look forward to revising our manuscript accordingly.
> > > > >
> > > > > [1] Bai, Yuntao, et al. "Training a helpful and harmless assistant with reinforcement learning from human feedback." arXiv preprint arXiv:2204.05862 (2022).
> > > > > [2] Touvron, Hugo, et al. "Llama 2: Open foundation and fine-tuned chat models." arXiv preprint arXiv:2307.09288 (2023).
> > > > > [3] Rafailov, Rafael, et al. "Direct preference optimization: Your language model is secretly a reward model." arXiv preprint arXiv:2305.18290 (2023).

---

### Official Review · Reviewer_iHby · 2023-11-01

**Soundness:** 3 good
**Presentation:** 3 good
**Contribution:** 4 excellent
**Rating:** 8
**Confidence:** 3

**Summary:**

This paper introduces a novel model alignment technique to user preferences. The authors have developed a simplified tuning method for point-wise preference data as well as human demonstration.

**Strengths:**

1. Detailed background and preliminaries section which serves as a refresher of the main LLM methodologies. This serves as a solid base and leads very well to the proposed methodology.
2. Detailed mathematical explanation of the concept.

**Weaknesses:**

The experiments section is not very detailed. Expanding the methodology to more datasets would be nice.

**Questions:**

None

---

> ### Author Response · Authors · 2023-11-19
> **Author Response**
>
> Thank you for the valuable review! We appreciate your positive assessment and will address your concerns in the following.
>
> **Q1:** "The experiments section is not very detailed. Expanding the methodology to more datasets would be nice."
>
> **A1:** We have added two more datasets *QA-feedback* and *red-team* in the revision. The former dataset is a point-wise dataset with binary labels, and the second dataset is a point-wise dataset with continuous labels. The empirical results are presented in Table 1 and Table 2 in the updated manuscript. Please refer to the manuscript for empirical results and detailed discussions.
>
> In particular, on the *red-team* dataset, we give an example of expanding the methodology of ULMA, i.e., using a hybrid objective formulation for different types of data, to handle point-wise dataset with continuous labels. Specifically, *red-team* is a point-wise dataset on LLM's robustness to red teaming attacks. It consists of samples scoring from 0 to 4, among which those rated 0 can be considered as high quality demonstration data. To better exploit these data, we introduce a hybrid loss by adding SFT loss of the samples rated 0 to the original MSE loss of all samples.

---

### Official Review · Reviewer_UxYc · 2023-11-01

**Soundness:** 3 good
**Presentation:** 4 excellent
**Contribution:** 3 good
**Rating:** 5
**Confidence:** 4

**Summary:**

In this paper, the authors propose a unified language model alignment approach. Their main idea is to address point-wise human preference. I like the idea of studying point-wise human preference. My main concern is that in many cases, there can be a mapping function between pair-wise preference and point-wise preference.

**Strengths:**

1. It is interesting to study the alignments on the point-wise human preference.
2. It is great to compare the existing approach.
3. Releasing more datasets is always great for the community.

**Weaknesses:**

1. In many cases, there can be a mapping function between pair-wise preference and point-wise preference. The authors do not discuss these cases.
2. It would be great to have more experimental results in terms of more LLM-based tasks.
3. There is no significance test in the tables.

**Questions:**

I like the idea of studying point-wise human preference. However, one essential issue is that in many cases, there can be a mapping function between pair-wise preference and point-wise preference. For example, from pair-wise -> point-wise: you can directly enumerate how many positive preferences have been received for each document, and then rank the documents according to the numbers and assign a ranking score to each document. Or, a simpler way is to use the number of (positive num – negative num) preferences as the point-wise preference. Therefore, to verify the idea of studying the point-wise human preference. Similarly, point-wise -> pair-wise, one document with higher scores can receive the positive preference. One must prove that these rule-based methods can not work well for the LLM. Also, to test the performance of an LLM, there are many evaluation metrics and many LLM-based tasks. Therefore, I expect the authors to test the LLM for more tasks and metrics. Also, more LLMs are expected. For the reported tables, many numbers are quite close, and it is necessary to have a significance test to see whether the proposed method is better (or you can report the mean and std for multiple runs). Overall, I like this idea, but this version may not be ready for publication. If you can answer the above question or point out my misunderstanding, I will be happy to raise my score.

---

> ### Author Response · Authors · 2023-11-19
> **Author Response**
>
> Thank you for the insightful review! We acknowledge your valuable suggestions and concerns, which we will address in the following.
>
> **Q1:** "In many cases, there can be a mapping function between pair-wise preference and point-wise preference"
>
> **A1:** Admittedly, there exists some relation between pair-wise data and point-wise data as you have suggested, but we shall note that it is **not** a one-to-one correspondence, and the transformation may suffer from information loss. Now we explicate the above point in two folds.
>
> (i) "Pair-wise to point-wise" transformation may lose some pair-wise comparison information. For example, consider two pair-wise datasets $\mathcal D_1=\\{x_1\succ x_2, x_2\succ x_3, x_3\succ x_4\\}$ and $\mathcal D_2=\\{x_1\succ x_3, x_3\succ x_2, x_2\succ x_4\\}$. Obviously, the two datasets differ from the pair-wise relation between $x_2$ and $x_3$. These two datasets will be transformed to the same point-wise dataset $\mathcal D_3=\\{(x_1,1), (x_2,0), (x_3,0), (x_4,-1)\\}$, in which $x_2$ and $x_3$ have the same score. In this case, the transformation drops the relation between $x_2$ and $x_3$, which incurs loss in information.
>
> (ii) "Point-wise to pair-wise" transformation is unsuitable to some dataset where no comparison can be made, e.g., all samples have the same score or there is only one sample for some prompt. One example is the *red-team* dataset, which only has one sample for each prompt. This dataset cannot be transformed into a pair-wise dataset, and pair-wise methods such as RLHF and DPO are inapplicable. Please see the revised manuscript for more detailed descriptions and empirical results.
>
> Therefore, the conversion between pair-wise and point-wise datasets is not always possible, and may suffer from information loss in many cases. In the revised manuscript, we have compared the performance of point-wise DPO and point-wise DPO on pair-wise dataset *Golden HH* and point-wise dataset *QA-feedback*, respectively. We observe that pair-wise DPO performs slightly worse than point-wise DPO on point-wise dataset, and vice versa, which supports our above claim.
>
> In addition to the above difference, we shall emphasize that **our investigation on point-wise dataset has motivated the design of more flexible tuning method**. Specifically, the absolute labeling on point-wise dataset motivates us to treat different types of samples (e.g., high quality positive data and noisy negative data) in different ways, which results in the ULMA framework. Our investigation on point-wise dataset and proposed new dataset may motivate more elaborate algorithmic design for point-wise preference learning.
>
> **Q2:** "more experimental results in terms of more LLM-based tasks"
>
> **A2:** We have added two more datasets *QA-feedback* and *red-team* in the revision. The former dataset is a point-wise dataset with binary labels, and the second dataset is a point-wise dataset with continuous labels. In particular, *red-team* only has a single answer for each prompt and no comparison can be made, hence it cannot be transformed into a pair-wise dataset and pair-wise methods such as RLHF and DPO are inapplicable. The empirical results are presented in Table 1 and Table 2 in the updated manuscript. Please refer to the manuscript for more details.
>
> **Q3:** "There is no significance test in the tables"
>
> **A3:** In the revised manuscript, for each  harmful/helpful score presented in Table 1-3, we report its 95-percent confidence interval after repeating the model training procedure for three times beyond its mean value. The results show that, on the *red-team* and *QA-feedback* datasets, ULMA performs **significantly** better than other methods; on the *HH* and *Golden HH* datasets, though the confidence intervals are overlapped, the mean value of ULMA is still better. Please refer to the manuscript for detailed empirical results.

---

### Official Review · Reviewer_veNV · 2023-11-01

**Soundness:** 3 good
**Presentation:** 3 good
**Contribution:** 3 good
**Rating:** 6
**Confidence:** 4

**Summary:**

Language model alignment is a significant technique to align inference output to human preference and help performance improvement. Currently, alignment mainly involves two steps: supervised fine-tuning with designed instructions and then preference learning with pair-wise samples such as RLHF and DPO method. However, most of the existing preference data in the world are not just pair-wise but more fine-grained, i.e., preference data are voted by scores. In this paper, the authors propose a new DPO method to align LLM with point-wise preference data. Standing on the proposed point-wise DPO method, they incorporate supervised fine-tuning, unifie the whole alignment framework, and solve it as a one-step alignment problem. In their experiments, they compare with RLHF and vanilla DPO and validate the effectiveness of their proposed framework by achieving lower perplexity scores and higher preference scores.

**Strengths:**

* Originality: Several existing works to align LLM outputs to human preference have been proposed, such as RLHF and DPO. Standing on DPO, this paper devises a new approach for point-wise preference data to make alignments. Besides, they unify the alignment framework with supervised fine-tuning stage. These two contributions enhances paper’s strength on originality.
* Quality: Numbers in the experiments are solid and look promising, especially the improvements in complexity and preference score (harmful) compared to baseline RLHF.
* Clarity: The presentation in this paper is easy to follow and well-organized.
* Significance: A typical way to do preference learning is to treat generated samples with pair-wise binary relation which losses the granular information on voting scores, rankings, or preference levels. To fill the gap, this paper proposes a new DPO method to align LLM with point-wise preference data. They study the gradients between supervised fine-tuning and their proposed method then propose a novel unified framework to learn human preference. Empirically, their results validate the framework’s effectiveness and show the significance of this work.

**Weaknesses:**

* Though the experimental results look promising to demonstrate framework’s effectiveness, more human preference datasets to align LLM should be included, such as datasets provided and used in [1] and [2].
* The proposed framework should be able to be generalized to more complex metrics (such as the discussion to handle continuous labels) but the datasets used in the experiment are only in binary classes, which is not enough to support the capability of its generalization.
* The generalization to other metrics with positive and negative samples needs further description in details.

[1] https://arxiv.org/abs/2112.09332

[2] https://proceedings.mlr.press/v162/ethayarajh22a.html

**Questions:**

* What’s the objective loss of ULMA for continuous preference labels?
* In this case, how does the framework deal with positive and negative samples?

---

> ### Author Response · Authors · 2023-11-19
> **Author Response**
>
> Thank you for the helpful review! We now address your concerns and suggestions as follows.
>
> **Q1:** "more human preference datasets to align LLM should be included"
>
> **A1:** We have added two more datasets *QA-feedback* and *red-team* in the revision. The former dataset is a point-wise dataset with binary labels, and the second dataset is a point-wise dataset with continuous labels. The new empirical results are presented in Table 1 and Table 2 (together with the original results on *HH* and *Golden HH*). Please refer to the revised manuscript for detailed descriptions and empirical results.
>
> **Q2:** "The proposed framework should be able to be generalized to more complex metrics (such as the discussion to handle continuous labels)... the dataset is not enough to support the capability of its generalization"
>
> **A2:** In fact, the core concept of our proposed ULMA framework, i.e., using a hybrid objective formulation for different types of data, is capable of generalizing to more general metrics. On the new *red-team* dataset in the revision, we provide an example of how ULMA can be generalized to handle continuous labels.
>
> Specifically, *red-team* is a point-wise dataset on LLM's robustness to red teaming attacks, which consists of samples scoring from 0 to 4. The scores are rated by human and indicate how successful the attacks are, among which those rated 0 can be considered as high quality demonstration data. To better exploit these data, we introduce a hybrid loss by adding SFT loss of the samples rated 0 to the original MSE loss of all samples.
>
> In experiment, we observe that ULMA performs better than Unlikelihood and point-wise RLHF on *red-team* (note that pair-wise methods such as RLHF and DPO cannot be applied to this dataset since each prompt only has a single sample), which supports the capability of ULMA's generalization to more complex metrics.
>
> **Q3:** "The generalization to other metrics with positive and negative samples needs further description in details"
>
> **A3:** Good suggestion! We have detailedly discussed how to generalize ULMA to datasets with continuous labels in Section 4.3 in the revised manuscript (at the end of page 6). Specifically, for these datasets, there is no direct separation of positive and negative samples. To apply ULMA, we can specify some sample as high quality data (e.g., the most helpful or harmless ones) and treat these high quality samples as "positive" demonstration data. Then we can adopt a hybrid loss by combining the SFT loss on these demonstration data with the original loss (e.g., MSE) on preference data. A concrete example on the *red-team* dataset can be seen in **A2**.
>
> **Q4:** "What’s the objective loss of ULMA for continuous preference labels? In this case, how does the framework deal with positive and negative samples?"
>
> **A4:** The objective loss for continuous preference labels is a hybrid of SFT loss on high quality demonstration data and the original loss (e.g., MSE) on preference data. In this case, some samples in the dataset (e.g., the most helpful or harmless ones) are treated as the "positive" demonstration data and applied to the SFT loss. An example on the *red-team* dataset is discussed and empirical evaluated in the revised manuscript. Please refer to our response in **A2** and **A3** for more details.

---

> > ### Comment · Reviewer_veNV · 2023-11-20
> >
> > Thanks for addressing my questions and adding more results! Two new datasets and their results look great and promising to me. Especially, the discussion about how this framework extends to continuous label spaces is important. I will raise my score to 6 and support this paper to be accepted.

---

### Author Response · Authors · 2023-11-21
**Eagerly look forward to knowing your update**

Respected Reviewers,

The discussion phase is drawing to a close. We eagerly look forward to knowing your update after the initial author response.

We are wondering whether your concerns have been well addressed. If you have any additional questions, it would be great if you could let us know. We are readily prepared to address them.

---

### Meta-Review · Area_Chair_9zti · 2023-12-06

**Metareview:**

The paper presents a novel approach to align language models with point-wise preference data, integrating supervised fine-tuning and preference learning into a unified framework. Strengths include the originality of handling point-wise preference data and unifying alignment frameworks, as well as solid experimental results showing improvements over existing methods. Weaknesses involve limitations in generalization to more complex metrics and a need for broader dataset application. Some concerns about the justification of designed components, remain partially unresolved.

**Justification For Why Not Higher Score:**

The paper is not recommended for a higher score primarily due to unresolved issues raised by reviewers UxYc and 33oF. While the authors made significant improvements in response to feedback, there remain gaps in the generalization to various metrics and datasets. Moreover, the clarity and justification of certain key components require further elaboration.

**Justification For Why Not Lower Score:**

Despite its limitations, the paper merits acceptance at its current score rather than a lower one. The research presents significant advancements in model alignment with point-wise preference data and demonstrates empirical improvements over existing methods. The issues raised, while pertinent, do not significantly detract from the overall value and novelty of the work.

---

### Decision · Program_Chairs · 2024-01-16

Reject